# A Unified View of Masked Image Modeling

**Zhiliang Peng**[*]                                                    *pengzhiliang19@mails.ucas.ac.cn*
*University of Chinese Academy of Sciences*

**Li Dong**                                                             *lidong1@microsoft.com*
*Microsoft Research*

**Hangbo Bao**                                                         *t-habao@microsoft.com*
*Microsoft Research*

**Furu Wei**[†]                                                        *fuwei@microsoft.com*
*Microsoft Research*

**Qixiang Ye**[†]                                                      *qxye@ucas.ac.cn*
*University of Chinese Academy of Sciences*

**Reviewed on OpenReview:** *https://openreview.net/forum?id=wmGlMhaBe0*

## Abstract

Masked image modeling has demonstrated great potential to eliminate the label-hungry problem of training large-scale vision Transformers, achieving impressive performance on various downstream tasks. In this work, we propose a unified view of masked image modeling after revisiting existing methods. Under the unified view, we introduce a simple yet effective method, termed as MASKDISTILL, which reconstructs normalized semantic features from teacher models at the masked positions, conditioning on corrupted input images. Experimental results on image classification and semantic segmentation show that MASKDISTILL achieves comparable or superior performance than state-of-the-art methods. When using the huge vision Transformer and pretraining 300 epochs, MASKDISTILL obtains 88.3% fine-tuning top-1 accuracy on ImageNet-1k (224 size) and 58.8% semantic segmentation mIoU metric on ADE20k (512 size). The code and pretrained models will be available at https://aka.ms/unimim.

## 1 Introduction

In recent years, Transformer architectures have shown promising results in the natural language processing field (Vaswani et al., 2017) and computer vision field (Dosovitskiy et al., 2020). Transformer, in the process of scaling up, is easy to overfit the small datasets and tends to demand more and more data. In NLP, self-supervised pretraining methods based on language modeling (Radford & Narasimhan, 2018; Devlin et al., 2019; Dong et al., 2019), have successfully addressed this problem. Motivated by masked language modeling, BEIT (Bao et al., 2022) proposes masked image modeling (MIM) to relieve the label-hungry problem of vision Transformers (ViT; Dosovitskiy et al. 2020), which shows impressive results in learning visual representations.

MIM is conceptually simple: models accept the corrupted input image and predict the target of the masked content. Take the pioneering work BEIT (Bao et al., 2022) as an example, the encoder accepts corrupted image patches as input and then predicts the corresponding discrete visual tokens from the tokenizer (Ramesh et al., 2021) at the masked positions. After that, the main difference between previous work lies in the architecture design (He et al., 2022; Chen et al., 2022a) and reconstruction targets (He et al., 2022; Liu et al., 2022b; Wei et al., 2021; 2022a; Baevski et al., 2022).

---

[*] Contribution during internship at Microsoft Research. [†] Corresponding authors.

In this work, we provide a unified view of masked image modeling, as illustrated in Equation 1 and Figure 1: a teacher model, a normalization layer, a student model, a MIM head, and a loss function. According to it, we conduct a systemic comparison of the recent MIM works and present it in Table 1. The most significant difference is the teacher model selection, *e.g.*, pixel values, tokenizers, pretrained models, and the momentum updated teacher.

Under this unified view, we induce a simple yet effective method, named MASKDISTILL. As shown in Figure 1, the ingredients of MASKDISTILL contain a teacher model based on CLIP (Radford et al., 2021), a fully-connection layer MIM head, layer normalization for target feature, and the Smooth-$\ell_1$ loss function. Compared to existing methods in Table 1, MASKDISTILL is loyal to the most straightforward design, but shows impressive results. Compared to knowledge distillation, MASKDISTILL pays more attention to extrapolating the masked patches rather than mimicking the target features.

We conduct MIM pretraining on ImageNet-1k (Russakovsky et al., 2015) for base-, large- and huge-size ViTs. After that, we evaluate pretraining models on downstream visual tasks, image classification on ImageNet-1k, and semantic segmentation on ADE20k (Zhou et al., 2019). With the large-size CLIP teacher, MASKDISTILL using ViT-H/14 can achieve 88.3% accuracy on ImageNet-1k and 58.8 mIoU on ADE20k, by pretraining 300 epochs.

The contributions of this study are summarized as follows:

- We provide a unified view of masked image modeling: a teacher model, a normalization layer, a student model, a MIM head, and a loss function.

- We propose a simple yet effective method, termed as MASKDISTILL.

- We conduct extensive experiments on downstream tasks including ImageNet fine-tuning and semantic segmentation. Experimental results show that the proposed approach improves performance across various settings.

## 2    A Unified View of Masked Image Modeling

In this section, we provide a unified view of the masked image modeling (MIM) task: a teacher model $\mathcal{T}$, a normalization layer $\mathcal{N}$, a student model $\mathcal{S}$, a MIM head $\mathcal{H}$, and an objective function $\mathcal{L}$ that measures the distance between the representation of the teacher model $\mathcal{T}$ and that of the student model $\mathcal{S}$. The pretraining task can be unified as:

$$\text{MIM} = \mathcal{L}(\mathcal{N}(\mathcal{T}(I_{\text{full}})), \mathcal{H}(\mathcal{S}(I_{\text{masked}}))) \tag{1}$$

where $I_{\text{full}}$ and $I_{\text{masked}}$ denote the full (original) image and the masked image respectively. According to Equation 1, we summarize the recent popular MIM works in Table 1.

1) *Teacher models* $\mathcal{T}$. According to the semantic information of target, we split them into two groups: *low-level* and *high-level* target. For the low-level target, ViT (Dosovitskiy et al., 2020), MAE (He et al., 2022), SimMIM (Liu et al., 2022b), ConvMAE (Gao et al., 2022), HiViT (Zhang et al., 2022) and GreenMIM (Huang et al., 2022) utilize the original or normalized pixels as the MIM target. MaskFeat (Wei et al., 2021) introduces the feature descriptor HOG (Dalal & Triggs, 2005) as the regression target. And Ge$^2$-AE regresses pixel and frequency from 2D-Discrete Fourier Transform in parallel. As for high-level target, BEIT (Bao et al., 2022), CAE (Chen et al., 2022a), SplitMask (El-Nouby et al., 2021), PeCo (Dong et al., 2021) and BEIT v2 (Peng et al., 2022) predict the discrete tokens (instantiated as code in the visual tokenizer (Ramesh et al., 2021; Esser et al., 2021; Peng et al., 2022)). MaskFeat (Wei et al., 2021) proposes to directly regress the pretrained model (*e.g.*, DINO (Caron et al., 2021) and DeiT (Touvron et al., 2020)). MVP (Wei et al., 2022a) extends the pretrained model to the multimodal pretrained model CLIP (Radford et al., 2021). Moreover, following the BYOL paradigm (Grill et al., 2020), data2vec (Baevski et al., 2022), MSN (Assran et al., 2022), ConMIM (Yi et al., 2022), SIM (Tao et al., 2022) and BootMAE (Dong et al., 2022) construct the regression target from the momentum updated teacher to boost itself online.

Table 1: Systemic comparisons of masked image modeling methods from a unified view.

| Methods | Teacher $\mathcal{T}$ | Student $\mathcal{S}$ | MIM Head $\mathcal{H}$ | Normalization $\mathcal{N}$ | Loss Function $\mathcal{L}$ |
|---|---|---|---|---|---|
| *Low-level pixel / feature* | | | | | |
| ViT (Dosovitskiy et al., 2020) | Pixel | ViT | FC | / | N/A |
| MAE (He et al., 2022) | Pixel | ViT | Decoder | LayerNorm | $\ell_2$ |
| SimMIM (Liu et al., 2022b) | Pixel | Swin | FC | / | $\ell_1$ |
| MaskFeat (Wei et al., 2021) | HOG | ViT | FC | / | $\ell_2$ |
| Ge$^2$-AE (Liu et al., 2022a) | Pixel&Frequency | ViT | Decoders | / | $\ell_2$ |
| ConvMAE (Gao et al., 2022) | Pixel | Hybrid ViT | Decoder | LayerNorm | $\ell_2$ |
| HiViT (Zhang et al., 2022) | Pixel | HiViT | Decoder | LayerNorm | $\ell_2$ |
| GreenMIM (Huang et al., 2022) | Pixel | Swin | Decoder | LayerNorm | $\ell_2$ |
| *High-level feature* | | | | | |
| BEiT (Bao et al., 2022) | dVAE | ViT | FC | / | CrossEntropy |
| CAE (Chen et al., 2022a) | dVAE | ViT | Decoder | / | CrossEntropy |
| SplitMask (El-Nouby et al., 2021) | dVAE | ViT | Decoder | / | InfoNCE&CrossEnt. |
| PeCo (Dong et al., 2021) | VQGAN | ViT | FC | / | CrossEntropy |
| BEiT v2 (Peng et al., 2022) | VQ-KD | ViT | FC | / | CrossEntropy |
| MaskFeat (Wei et al., 2021) | DINO | ViT | FC | $(\ell_2)$ | Cosine |
| MVP (Wei et al., 2022a) | CLIP | ViT | FC | $(\ell_2)$ | Cosine |
| MILAN (Hou et al., 2022) | CLIP | ViT | Decoders | $\ell_2$-Norm | $\ell_2$ |
| MimCo (Zhou et al., 2022) | MoCov3 | ViT | FC | / | InfoNCE |
| data2vec (Baevski et al., 2022) | EMA | ViT | FC | LayerNorm | Smooth-$\ell_1$ |
| MSN (Assran et al., 2022) | EMA | ViT | FC | / | CrossEntropy |
| SIM (Tao et al., 2022) | EMA | ViT | Decoder | BatchNorm | UniGrad loss |
| SdAE (Chen et al., 2022b) | EMA | ViT | Decoder | LayerNorm | Cosine |
| ConMIM (Yi et al., 2022) | EMA | ViT | FC | BatchNorm | InfoNCE |
| ExtreMA (Wu et al., 2022) | EMA | ViT | CrossAtt | LayerNorm | Cosine |
| BootMAE (Dong et al., 2022) | EMA&Pixel | ViT | Decoders | LayerNorm | $\ell_2$ |
| **MaskDistill (Ours)** | CLIP | ViT | FC | LayerNorm | Smooth-$\ell_1$ |

2) *Student models $\mathcal{S}$.* MIM task is suitable for the models root in attention interaction, like ViT (Dosovitskiy et al., 2020), Swin Transformers (Liu et al., 2022b), and some variants (Gao et al., 2022; Zhang et al., 2022). Because backbone architecture is not the primary focus of this study, we choose the vanilla ViT (Dosovitskiy et al., 2020) as the analytical anchor.

3) *MIM Heads $\mathcal{H}$.* BEiT (Bao et al., 2022) uses a simple fully-connection (FC) layer as the task head to generate prediction at the masked positions. MAE (He et al., 2022) introduces a decoder to decouple the masked prediction task from the encoder. In fact, the aim of the decoder in MAE is still to predict the target pixel at the masked positions. Therefore, we consider the decoder as a MIM head in Table 1. And this decoupling decoder is adopted by many recent works (Liu et al., 2022a; Gao et al., 2022; Zhang et al., 2022; Chen et al., 2022a; El-Nouby et al., 2021; Tao et al., 2022; Dong et al., 2022).

4) *Normalization Layers $\mathcal{N}$.* MAE (He et al., 2022) also introduces per-patch normalized pixels (*i.e.*, layer normalization without affine transformation) as the target to boost local pixels contrast, resulting in better performance. Meanwhile, normalization is usually applied for avoiding feature collapse in methods based on contrastive learning (Grill et al., 2020; Chen et al., 2020). Similarly, EMA-based MIM methods (Tao et al., 2022; Baevski et al., 2022; Yi et al., 2022) adopt various normalization methods to stabilize training as well as boost performance. There is no collapse issue when the teacher is pixels or frozen models by default.

5) *Loss functions $\mathcal{L}$.* When the target is pixel or feature, $\ell_1$ or $\ell_2$ losses are appropriate for feature regression. When the target is discrete tokens, the cross entropy loss is the primary choice. Notably, after applying layer normalization, the variance of target feature rises, resulting in volatile loss, whereas Smooth-$\ell_1$ loss is a trade-off between $\ell_1$ and $\ell_2$, performing more stable. Of course, cosine similarity loss is also an alternative choice.

From Table 1, one can find that the main difference is the teacher models: pixel, momentum-updated teachers, and pretrained models. Pixel is easy to access but struggles with low-level semantic knowledge. Momentum-updated teachers do not need extra models or datasets but tend to suffer from the collapse

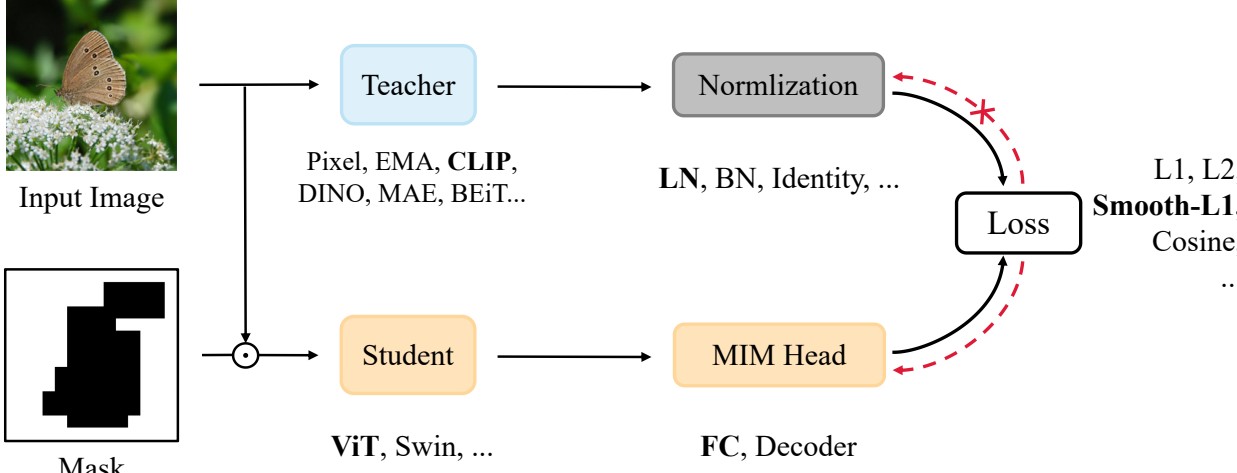

Figure 1: Unified view of the masked image modeling framework. The **bold** text denotes the default ingredients of MASKDISTILL.

issue. Pretrained models are off-the-shelf and contain more rich semantic information than pixels, but how to prepare a high-quality teacher model is an essential problem.

## 3    Masked Distillation

Knowledge distillation (Hinton et al., 2015) has shown to be a promising approach for compressing a large model (referred to as the teacher model) into a small model (referred to as the student model), which utilizes much fewer parameters and computations while attaining comparable results on downstream tasks.

Based on the unified view, we offer a simple yet effective method, named MASKDISTILL, to distill a student model in a masked image modeling fashion. However, our purpose is not to compress the teacher model $\mathcal{T}$ into the student model $\mathcal{S}$, but to boost $\mathcal{S}$ to outperform $\mathcal{T}$. We instantiate the student model $\mathcal{S}$ as ViT (Dosovitskiy et al., 2020) for comparison with others.

Specially, given the input image $\boldsymbol{x} \in \mathbb{R}^{H \times W \times C}$, where $(H, W)$ is the resolution and $C$ is the number of image channels, the student $\mathcal{S}$ first divides $\boldsymbol{x}$ into $N$ non-overlapping patches $\{\boldsymbol{x}_i^p\}_{i=1}^N$ and then linear projects it into patch embeddings $\{\boldsymbol{e}_i^p\}_{i=1}^N$. Following that, we select roughly 40% of the image patch embeddings to be masked, in a block-wise strategy (Bao et al., 2022). Denoting the masked position set as $\mathcal{M}$, we use a shared learnable embedding $\boldsymbol{e}_M$ to replace the original patch embeddings $\boldsymbol{e}_i^p$ if $i \in \mathcal{M}$. After that, we get the masked sequence:

$$\boldsymbol{e}_i^{\mathcal{M}} = \delta(i \in \mathcal{M}) \odot \boldsymbol{e}_M + (1 - \delta(i \in \mathcal{M})) \odot \boldsymbol{e}_i^p, \tag{2}$$

where $\delta(\cdot)$ is the indicator function. Subsequently, we prepend a learnable class token $\boldsymbol{e}_{\text{CLS}}$ and add the learnable positional embeddings, and then feed those into stacked transformer blocks. Lastly, a masked image modeling head (usually instantiate as a fully-connected layer) is applied for predicting feature $\mathbf{O} \in \mathbb{R}^{(N+1) \times D}$, where $D$ is the dimension of target features.

Given a pretrained teacher model $\mathcal{T}$, like DINO (Caron et al., 2021) and CLIP (Radford et al., 2021), the same image $\boldsymbol{x}$ is fed into $\mathcal{T}$ to get the target feature $\{t_i\}_{i=1}^N$ patch-to-patch. To ensure that the output resolution of $\mathcal{S}$ and $\mathcal{T}$ is the same, the input resolution for $\mathcal{T}$ should be adjusted. Finally, the training objective of MASKDISTILL can be formulated as:

$$\mathcal{L}_{\text{MASKDISTILL}} = -\sum_{i \in \mathcal{M}} \log(t_i(x)|\boldsymbol{x}_i^p) = \frac{1}{|\mathcal{M}|} \sum_{i \in \mathcal{M}} \text{Smooth-}\ell_1(o_i, LN(t_i)), \tag{3}$$

where $LN$ is the layer normalization without affine transformation.

Table 2: Fine-tuning results on ImageNet-1K and ADE20k. 'PT Epochs' denotes the pretraining epochs. 'Rel. Pos.' means relative positional embeddings.

| Methods | Teacher Model | Data | PT Epochs | Rel. Pos. | Classification Top-1 Acc (%) | Segmentation mIoU (%) |
|---|---|---|---|---|---|---|
| *Base-size models (ViT-B/16)* | | | | | | |
| BEiT (Bao et al., 2022) | DALL-E | 250M | 800 | ✗ | 83.2 | 45.6 |
| MAE (He et al., 2022) | Pixel | / | 1600 | ✗ | 83.6 | 48.1 |
| CAE (Chen et al., 2022a) | DALL-E | 250M | 1600 | ✗ | 83.9 | 50.2 |
| SdAE (Chen et al., 2022b) | EMA | / | 300 | N/A | 84.1 | 48.6 |
| SIM (Tao et al., 2022) | EMA | / | 1600 | N/A | 83.8 | N/A |
| MaskFeat (Wei et al., 2021) | HOG | / | 1600 | N/A | 84.0 | N/A |
| PeCo (Dong et al., 2021) | VQGAN | IN-1k | 300 | N/A | 84.1 | 46.7 |
| PeCo (Dong et al., 2021) | VQGAN | IN-1k | 800 | N/A | 84.5 | 48.5 |
| data2vec (Baevski et al., 2022) | EMA | / | 800 | ✗ | 84.2 | N/A |
| CLIP (Radford et al., 2021) | Text | / | / | ✗ | 84.9 | 51.1 |
| MVP (Wei et al., 2022a) | CLIP-B | 400M | 300 | N/A | 84.4 | 52.4 |
| BEiT v2 (Peng et al., 2022) | VQ-KD | 400M | 1600 | ✓ | 85.5 | 53.1 |
| MASKDISTILL (ours) | CLIP-B | 400M | 300 | ✗ | 84.9 | 52.7 |
| MASKDISTILL (ours) | CLIP-B | 400M | 300 | ✓ | 85.0 | 53.8 |
| MASKDISTILL (ours) | CLIP-B | 400M | 800 | ✓ | 85.5 | 54.3 |
| *Large-size models (ViT-L/16)* | | | | | | |
| MaskFeat (Wei et al., 2021) | HOG | / | 1600 | N/A | 85.7 | N/A |
| MAE (He et al., 2022) | Pixel | / | 1600 | ✗ | 85.9 | 53.6 |
| CAE (Chen et al., 2022a) | DALL-E | 250M | 1600 | ✗ | 86.3 | 54.7 |
| data2vec (Baevski et al., 2022) | EMA | / | 1600 | ✗ | 86.6 | N/A |
| BEiT v2 (Peng et al., 2022) | VQ-KD | 400M | 1600 | ✓ | 87.3 | 56.7 |
| MILAN (Hou et al., 2022) | CLIP-B | 400M | 400 | ✗ | 86.7 | 55.3 |
| MASKDISTILL (ours) | CLIP-B | 400M | 300 | ✓ | 86.8 | 56.3 |
| MASKDISTILL (ours) | CLIP-B | 400M | 800 | ✓ | 87.1 | 56.5 |

Table 3: Fine-tuning results on ImageNet-1K and ADE20k. The teacher is CLIP ViT-L/14.

| Methods | Model Size | PT Epochs | Rel. Pos. | Classification Top-1 Acc (%) | Segmentation mIoU (%) |
|---|---|---|---|---|---|
| *Scaling up to larger teacher, CLIP ViT-L/14* | | | | | |
| MASKDISTILL (ours) | ViT-B/16 | 300 | ✓ | 85.3 | 54.3 |
| MASKDISTILL (ours) | ViT-L/16 | 300 | ✓ | 87.6 | 57.9 |
| MASKDISTILL (ours) | ViT-H/14 | 300 | ✓ | 88.3 | 58.8 |

## 4 Experiments

We perform pretraining and then evaluate fine-tuning performance on various downstream tasks, such as image classification and semantic segmentation. Moreover, we conduct ablation studies to compare the contributions of different design choices.

### 4.1 Setup

For all pretraining experiments, we only use the ImageNet-1k dataset (Russakovsky et al., 2015) contains 1.28M images. We adopt the block masking strategy to corrupt the input images for the student model, but keep the full images for the teacher, to construct the asymmetric informational bottleneck. All the teacher

Table 4: Robustness evaluation on ImageNet variants (Hendrycks et al., 2021b;a; Wang et al., 2019).

| Methods | ImageNet Adversarial | ImageNet Rendition | ImageNet Sketch |
|---|---|---|---|
| *ViT-B/16* | | | |
| MAE | 35.9 | 48.3 | 34.5 |
| BEiT v2 | **54.4** | 61.0 | 45.6 |
| MASKDISTILL | 53.3 | **64.4** | **47.3** |
| *ViT-L/16* | | | |
| MAE | 57.1 | 59.9 | 45.3 |
| BEiT v2 | **69.0** | 69.9 | 53.5 |
| MASKDISTILL | **69.0** | **75.3** | **56.9** |

Table 5: MASKDISTILL *vs* knowledge distillation. The teacher model is CLIP ViT-Base (Radford et al., 2021).

| Student Models | Mask Ratios | Pretaining Epochs | Classification Accuracy (%) |
|---|---|---|---|
| ViT-B/16 | 0 | 300 | **85.3** |
| | 40% | 300 | 85.0 (-0.3) |
| ViT-B/16 | 0 | 800 | 85.2 |
| | 40% | 800 | **85.5 (+0.3)** |
| ViT-L/16 | 0 | 300 | 85.4 |
| | 40% | 300 | **86.8 (+1.4)** |

model checkpoints are from the official publication. When utilizing CLIP ViT-L/14 as a teacher, we set the input image resolution to 196×196 for the teacher to match the number of patches with student ViT-B/16 or ViT-L/16. As for the student model, we use the ViT-Base/Large equipped relative positional embeddings following BEiT (Bao et al., 2022; Peng et al., 2022). For the pretraining setting, we mainly follow BEiT (Bao et al., 2022; Peng et al., 2022): batch size 2048, learning rate 1.5e-3, AdamW optimizer with weight decay 0.05, drop path 0.1 (0.2) for ViT-Base(large), block-wise mask 40%, epochs 300/800. More details can be found in Appendix.

**Evaluation.** We consider the popular evaluating protocol for image classification on ImageNet-1k dataset: *fine-tuning* top-1 accuracy. We adopt the BEiT (Bao et al., 2022) fine-tuning recipe: For ViT-Base, we fine-tune it for 100 epochs with 20 epochs warm-up, and use AdamW optimizer with weight decay 0.05, learning rate 5e-4, and decays in a cosine schedule, layer decay 0.65; For ViT-Large, we fine-tune it for 50 epochs with 5 epochs warm-up, layer decay 0.75. For ViT-Huge, we fine-tune it for 30 epochs with 5 epochs warm-up, layer decay 0.85. All the resolutions of input images are $224 \times 224$.

As for the semantic segmentation task, we evaluate the *mIoU* metric on ADE20K dataset (Zhou et al., 2019) with UperNet (Xiao et al., 2018) framework. The input image resolution for training and evaluating are $512 \times 512$. Remarkably, for the ViT-H/14 in Table 3, we convert it to ViT-H/16 for semantic segmentation task. Similarly, AdamW optimizer with weight decay of 0.05 is applied. Additionally, the training steps are 160K, and the batch size is 16. And we employ learning rate {5e-5, 8e-5, 1e-4}, layer decay 0.75 (0.85), drop path 0.1 (0.2) for ViT-Base (Large). More details can be found in Appendix.

## 4.2 Main Results

Table 2 reports the top-1 accuracy of some self-supervised methods on ImageNet-1k using ViT (Dosovitskiy et al., 2020) models. For ViT-base, MASKDISTILL with 800 epochs pretraining schedule obtains 85.5% top-1 accuracy, surpasses CLIP (Radford et al., 2021), MVP Wei et al. (2022a), data2vec (Baevski et al., 2022) and MaskFeat (Wei et al., 2021) by 0.6%, 1.1%, 1.3% and 1.5% respectively. And MASKDISTILL also achieves comparable performance with BEiT v2 (Peng et al., 2022) on ImageNet-1k but outperforms BEiT v2 by 1.2 mIoU on ADE20k. More comparison with BEiT v2 can be found in Section 4.4. When scaling up the student to ViT-Large, MASKDISTILL achieves 86.8% top-1 accuracy and 56.3 mIoU. Compared to the recently proposed MILAN (Hou et al., 2022), MASKDISTILL outperforms it by 1% on the semantic segmentation task under the less pretraining epochs.

In Table 3, we use the CLIP ViT-Large/14 checkpoint as the teacher model and pretrain student models for 300 epochs. One can see that MASKDISTILL can get consistent improvements compared to teacher CLIP ViT-Base/16. Remarkably, MASKDISTILL can reach 88.3% accuracy on ImageNet-1k and 58.8 mIoU on ADE20k by using the ViT-Huge backbone.

**Robustness evaluation.** Following MAE (He et al., 2022) and BEiT v2 (Peng et al., 2022), we test the robustness of MASKDISTILL on three ImageNet validation sets, *i.e.*, ImageNet-Adversarial (Hendrycks et al.,

Table 6: Fow-shot image classification on ImageNet-1k. We freeze the backbone and only learn the classifier during training.

| Methods | FT | Few Shot Numbers | | | | |
|---|---|---|---|---|---|---|
| | | k=2 | k=4 | k=8 | k=16 | k=32 |
| *ViT-B/16* | | | | | | |
| BEiT (Bao et al., 2022) | 83.2 | 1.7 | 3.0 | 5.0 | 7.0 | 8.9 |
| MAE (He et al., 2022) | 83.6 | 11.5 | 21.5 | 31.5 | 39.5 | 46.4 |
| CLIP (Radford et al., 2021) | 84.9 | 35.4 | 45.2 | 55.1 | 61.3 | 65.6 |
| BEiT v2 (Peng et al., 2022) | 85.5 | 36.6 | 47.6 | 56.3 | 63.0 | 67.6 |
| MASKDISTILL (ours) | 85.5 | 37.8 | 48.7 | 56.3 | 62.3 | 66.2 |

2021b), ImageNet-Rendition (Hendrycks et al., 2021a) and ImageNet-Sketch (Wang et al., 2019). In Table 4, both MAE and BEiT v2 pretrain 1600 epochs, while MASKDISTILL pretrains 800 epochs but achieves comparable or superior performance.

### 4.3 Comparison with Knowledge Distillation

In Table 5, we compare MASKDISTILL with knowledge distillation, which can be considered as a special case of MASKDISTILL where the mask ratio is 0 and loss is calculated on all patches. Knowledge distillation surpasses MASKDISTILL by 0.3% when the pretraining schedule is 300 epochs, but is inferior to MASKDISTILL by 0.3% when the pretraining schedule is 800 epochs. Remarkably, MASKDISTILL outperforms knowledge distillation by a significant gain when the student model scales up to large-size models. The commonly used teacher model is CLIP ViT-Base, which reaches 84.9% fine-tuning accuracy in terms of image classification on ImageNet-1k.

When the student is larger than the teacher, the student is easy to fully reconstruct the latent space of the teacher without information bottleneck. This is why ViT-L/16 obtains comparable performance with ViT-B/16 (85.4% *vs* 85.3% in Table 5). But in MASKDISTILL, under the condition of the corrupted input, the student is encouraged to extrapolate the masked patches, rather than mimicking features at visible patches.

### 4.4 Comparison with BEiT v2

In BEiT v2 (Peng et al., 2022), CLIP ViT-Base as the teacher model is responsible for distilling a vector quantized visual tokenizer, which provides the supervision for the subsequent MIM phase. But compared with MASKDISTILL, the quantized mechanism in BEiT v2 omits some fine-grained details from the teacher model. And these details are beneficial to the fast convergence of MASKDISTILL, *e.g.*, MASKDISTILL achieves comparable image classification performance with 800 epochs pretraining while BEiT v2 need to pretrain 1600 epochs, as demonstrated in Table 2. That is, MASKDISTILL can avoid the codebook collapse problem in the tokenizer training phase (Peng et al., 2022) and achieve comparable performance. Meanwhile, such fine-grained details as supervision enhance the robustness of MASKDISTILL, as shown in Table 4.

### 4.5 Few shot classification

In Table 6, we report the fewshot image classification results of BEiT (Bao et al., 2022), MAE (He et al., 2022), CLIP (Radford et al., 2021), BEiT v2 (Peng et al., 2022) and MASKDISTILL on ImageNet-1k (Russakovsky et al., 2015). We randomly choose k samples from each category and use them to learn a classifier, while keeping other parameters frozen during training. We use the entire validation set for evaluation. For each method, we use their public model weights and sweep a wide range of learning rates for a fair comparison. From Table 6, one can see that MASKDISTILL can consistently surpass the teacher under various settings. When k=2/4/8, MASKDISTILL can achieve the best fewshot performance. Moreover, we also find that fewshot performance is correlated with fine-tuning performance in terms of MIM, which is consistent with observation in contrastive learning (Ericsson et al., 2021).

Table 7: Comparison of the MIM time consuming and GPU memory.

| Method | Supervision | Average step time (s) | GPU memory (G) |
|---|---|---|---|
| *ViT-B/16, batchsize 64 on each GPU* | | | |
| MAE (He et al., 2022) | Normalized pixels | 0.358 | 10.6 |
| data2vec (Baevski et al., 2022) | EMA features | 0.636 | 13.5 |
| BEiT v2 (Peng et al., 2022) | VQ-KD tokens | 0.605 | 15.5 |
| MASKDISTILL | CLIP-B features | 0.487 | 12.5 |
| *ViT-L/16, batchsize 32 on each GPU* | | | |
| MAE (He et al., 2022) | Normalized pixels | 0.551 | 13.0 |
| data2vec (Baevski et al., 2022) | EMA features | 1.079 | 21.9 |
| BEiT v2 (Peng et al., 2022) | VQ-KD tokens | 0.769 | 21.8 |
| MASKDISTILL | CLIP-B features | 0.718 | 19.7 |

## 4.6 Comparison on Training Cost

In Table 7, We test all models under the same settings to compare the training cost. Average step time is calculated from (total training time) / (total steps). GPU memory is measured when the training phase is stable. MAE (He et al., 2022), data2vec (Baevski et al., 2022) and BEiT v2 (Peng et al., 2022) are evaluated by using their official codebase. Compared with using pixel values as reconstruction, other methods tend to spending more time on obtaining targets, including our MASKDISTILL. Compared with EMA-based data2vec and tokenizer-based BEiT v2, MASKDISTILL enjoys faster training time and lower GPU memory cost.

## 4.7 Ablation Studies

**Teacher models.** We collect some popular models to act as the teacher in MASKDISTILL, and pretrain a student model ViT-Base for 300 epochs in a MIM fashion. The performance of the teacher and student are shown in Table 8. From #1 to #6, where teacher models are CLIP and SLIP (Mu et al., 2021) trained on the image-text pair datasets (YFCC15M, CC3M, CC12M and private 400M) in a language-guided contrastive way, MASKDISTILL consistently boost the teacher model by 0∼3.3% accuracy. For #7, teacher model is ResNet, which is trained in the supervised way. Despite the gap in architecture, the student still enjoys the significant gain (83.5% *vs* 76.2%). From #8 to #9, teacher models SimCLR (Chen et al., 2020) and DINO (Caron et al., 2021) only use image data. MASKDISTILL boosts them by 1.6% and 0.9% respectively.

Comparing #1, #2 and #8 in Table 8, where the same dataset and training epochs are applied to teachers, students in #1 and #8 respectively achieve 83.8% and 84.1%, but the former using the text information and the later not, implying that the language-guided supervision is not essential. Moreover, comparing #1∼#5 and #9, both teacher and student in #9 trained on ImageNet-1k can reach comparable performance with those in #1∼#5, which further suggests that the extra language information is not the key.

From #10 to #12, we choose the model trained by MIM itself to act as the teacher model. We find that MASKDISTILL consistently outperform the corresponding teacher. However, comparing #9 with #10, where teacher can reach the same fine-tuning accuracy, students in #8 can obtain better performance in terms of fine-tuning accuracy and segmentation mIoU than those in #9, indicating that contrastive pretrained models tend to be the better but not the only solution.

**Loss functions & Normalization.** We compare MSE, cosine similarity and smooth-$\ell_1$ loss equipped with various normalization layers, then present the results in Table 9. From Table 9, one can see that smooth-$\ell_1$ loss equipped with LN can achieve better performance under the supervision of both DINO and CLIP, indicating that Normalization plays an important role in masked image modeling task.

**Target layer selection.** Usually, the deeper layer feature of a model is biased to the special task, *e.g.*, image-image contrastive learning in DINO and image-text contrastive learning in CLIP. But whether it

Table 8: Ablation studies on teacher models used in MASKDISTILL. For {CLIP, SLIP, SimCLR}[‡], the fine-tuning accuracy and model checkpoint are all from SLIP (Mu et al., 2021). For CLIP[*] and DINO, we use the official model checkpoint and follow BEiT (Bao et al., 2022) fine-tuning recipe to get the top-1 accuracy. The teacher models in all methods are *ViT-Base* model except ResNet-50 (He et al., 2016). The student model is ViT-Base and is pretrained for 300 epochs.

| | | Teacher Model $\mathcal{T}$ | | | Student Model $\mathcal{S}$ | |
| | Teacher | Data | Text | ImageNet (%) | ImageNet (%) | ADE20k (%) |
|---|---|---|---|---|---|---|
| #1 | CLIP[‡] | YFCC15M | ✓ | 80.5 | 83.8 (+3.3) | 47.4 |
| #2 | SLIP[‡] | YFCC15M | ✓ | 82.6 | 84.3 (+1.7) | 49.9 |
| #3 | SLIP[‡] | YFCC15M | ✓ | 83.4 | 84.6 (+1.2) | 50.8 |
| #4 | CLIP[‡] | CC3M | ✓ | 79.5 | 83.7 (+4.2) | 45.7 |
| #5 | CLIP[‡] | CC12M | ✓ | 82.1 | 84.1 (+2.0) | 48.3 |
| #6 | CLIP[*] | Private 400M | ✓ | 84.9 | **85.0** (+0.1) | **53.8** |
| #7 | ResNet | ImageNet-1k | ✗ | 76.2 | 83.5 (+7.3) | 46.9 |
| #8 | SimCLR[‡] | YFCC15M | ✗ | 82.5 | 84.1 (+1.6) | 49.4 |
| #9 | DINO | ImageNet-1k | ✗ | 83.6 | 84.5 (+0.9) | 50.4 |
| #10 | MAE | ImageNet-1k | ✗ | 83.6 | 84.3 (+0.7) | 49.3 |
| #11 | BEiT | ImageNet-1k | ✗ | 83.2 | 83.8 (+0.6) | 46.6 |
| #12 | BEiT v2 | ImageNet-1k | ✗ | 84.7 | **85.0** (+0.3) | 52.1 |

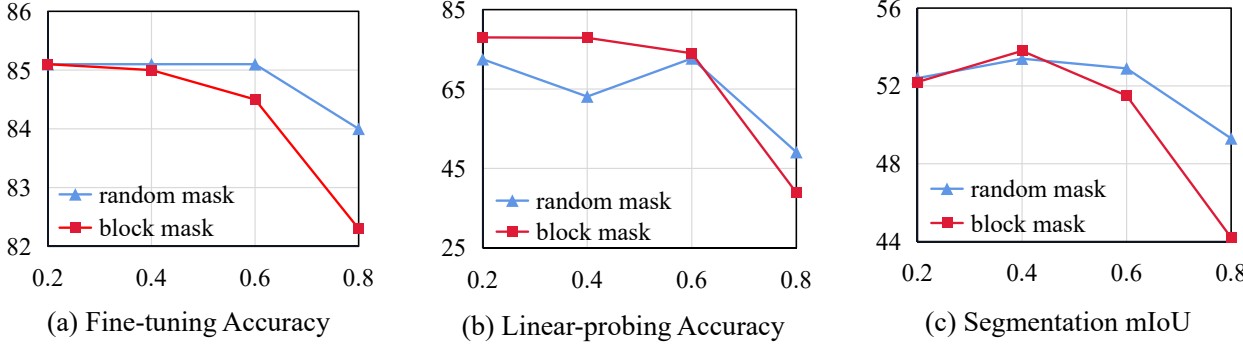

(a) Fine-tuning Accuracy
(b) Linear-probing Accuracy
(c) Segmentation mIoU

Figure 2: The block-wise mask *vs* random mask, under various mask ratios.

is beneficial for MASKDISTILL is not revealed. We conduct experiments on target feature from last layer, average of last 3 layers and average of last 6 layers. As shown in Table 10, the last layer's features are better for DINO teachers while the last 6 layers' features are better for CLIP teachers. Moreover, results on the segmentation task show that the last layer features as target are superior. Therefore, we choose the last layer feature as the default target feature for all experiments.

**Masked strategy.** For the masked strategy, we evaluate the block-wise (Bao et al., 2022) masked method and random masked method in Figure 2. The block-wise masked method performs better than random mask under low mask ratios, while worse than random mask under high mask ratios. Taking the three evaluation protocols (fine-tuning on ImageNet-1k, linear-probing ImageNet-1k, and semantic segmentation on ADE20k) into consideration, we choose the block-wise mask with 40% mask ratio as the final decision.

## 4.8 Analysis: MIM Enhances Shape Bias

We explore whether the masked image modeling methods can enhance the shape-biased ability or not. The fraction of correct decisions based on object shape is characterized as shape bias. Naseer et al. (2021) present

Table 9: Ablation study of loss functions and normalization layers. All models are pretrained for 300 epochs.

| $\mathcal{T}$ | $\mathcal{L}$ | Norm | ImageNet | ADE20k |
|---|---|---|---|---|
| | MSE | ✗ | 84.3 | 49.6 |
| DINO | Cosine | ($\ell_2$) | 84.5 | 49.6 |
| | Smooth-$\ell_1$ | LN | 84.5 | 50.4 |
| | MSE | ✗ | 84.6 | 52.8 |
| CLIP | Cosine | ($\ell_2$) | 84.9 | 52.9 |
| | Smooth-$\ell_1$ | BN | 84.9 | 53.1 |
| | Smooth-$\ell_1$ | LN | 85.0 | 53.8 |

Table 10: Ablation study of target feature selection in MASKDISTILL. All models are pretrained for 300 epochs.

| $\mathcal{T}$ | Target | ImageNet | ADE20k |
|---|---|---|---|
| | Last layer | 84.5 | 50.4 |
| DINO | Mean (last 3 layers) | 84.4 | 49.7 |
| | Mean (last 6 layers) | 84.3 | 49.8 |
| | Last layer | 85.0 | 53.8 |
| CLIP | Mean (last 3 layers) | 85.0 | 53.5 |
| | Mean (last 6 layers) | 85.1 | 53.4 |

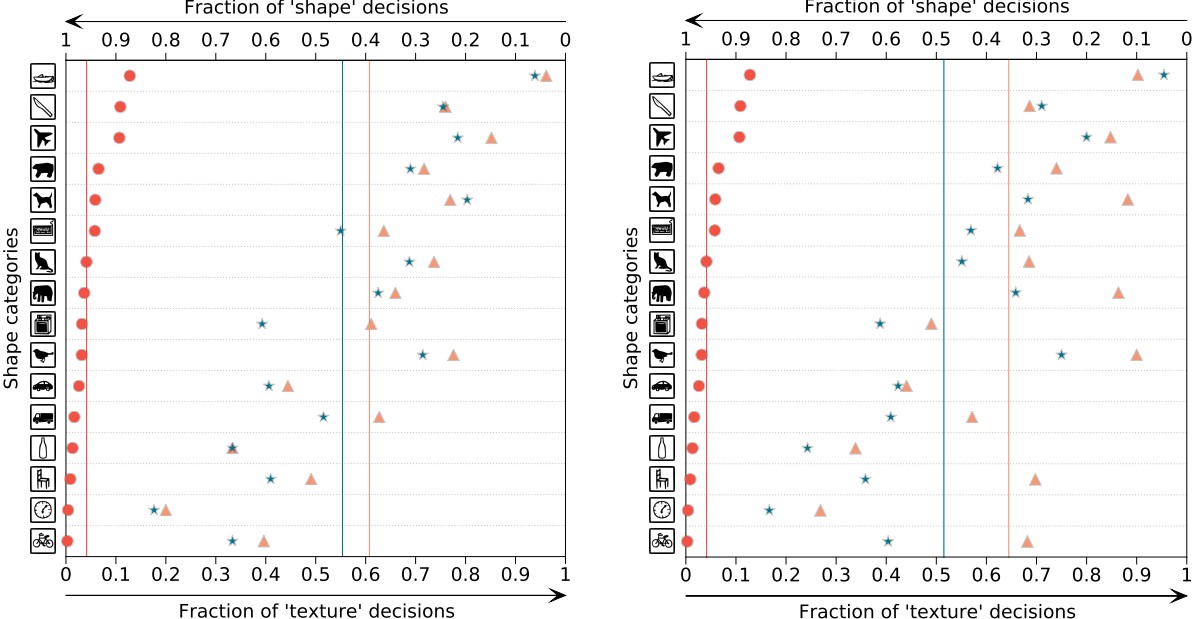

Figure 3: Shape-biased analysis under the teacher supervision of CLIP ViT-B/16 (**left**) and MAE ViT-B/16 (**right**). Circle, triangle and star denote humans, teachers and students, respectively. Vertical lines are the corresponding average values. Masked image modeling enhances the shape bias. (Best viewed in color)

that human usually is much more shape-biased compared with supervised classification models, such as convolutional networks, and vision Transformers. We evaluate the shape bias capacity on a stylized version of ImageNet (Naseer et al., 2021) by using the checkpoints fine-tuned on the original ImageNet-1k dataset. As shown in Figure 3, masked image modeling tends to promote the shape bias of the models. The results partially explains why MASKDISTILL generalizes better on ImageNet variants as shown in Table 4.

## 5 Related Work

**Masked image modeling.** Masked language modeling task root in Transformers has achieved great success in learning strong language representations in recent years (Devlin et al., 2019; Dong et al., 2019; Bao et al., 2020). Inspired by it, BEIT (Bao et al., 2022) proposes a mask-then-predict framework to recover discrete visual tokens (Ramesh et al., 2021), which shows the great potential of masked image modeling for the computer vision field. After that, various target supervision has been explored under the masked image modeling framework, such as original or normalized pixels (He et al., 2022; Dong et al., 2021; Liu et al.,

2022b; Gao et al., 2022; Liu et al., 2022a; Zhang et al., 2022; Huang et al., 2022), high-level features (Wei et al., 2021; 2022a; Peng et al., 2022; Zhou et al., 2022; Hou et al., 2022), and EMA-updated models (Baevski et al., 2022; Assran et al., 2022; Tao et al., 2022; Chen et al., 2022b; Yi et al., 2022; Wu et al., 2022; Dong et al., 2022). In this work, we decouple and analyze the components of the recent masked image modeling works, and then propose a simple yet effective paradigm for masked image modeling.

**Contrastive learning.** As a simple but effective self-supervised method, contrastive learning methods have ushered in rapid progress in recent years. The main idea is to enforce similarity over augmented views of an image and push the views augmented from other images away (Dosovitskiy et al., 2016; Wu et al., 2018; Hjelm et al., 2019; He et al., 2020; Chen et al., 2020), or to avoid model collapse after removing negative pairs (Grill et al., 2020; Chen & He, 2020; Chen et al., 2021; Caron et al., 2021). In the multimodal field, CLIP (Radford et al., 2021) and ALIGN (Jia et al., 2021) can learn image-language alignment representation, by grouping positive image-text pairs (an image and corresponding tag or caption) closer and separating negative image-text pairs. And SLIP (Mu et al., 2021) combines language supervision and image self-supervision to further boost the learned visual representations. In this work, we consider contrastive models as the target for masked image modeling.

**Knowledge distillation.** Knowledge distillation (Hinton et al., 2015) considers the output of the teacher model as the pseudo label to learn the student model. Such a strategy squeezes the potential of small models and brings impressive gains. After that, knowledge distillation is transferred to various tasks (Touvron et al., 2020; He et al., 2019; Yang et al., 2021) and domains (Jiao et al., 2020; Wang et al., 2020). Wei et al. (2022b) proposes that using the normalized feature from teacher fully distills a same size student. However, in this work, MASKDISTILL aims to reconstruct the corresponding teacher output at masked patches rather than mimicking the teacher's feature at each patch.

## 6    Conclusion and Limitations

We summarized the existing MIM works upon the proposed unified view: teacher models, student models, normalization layers and MIM heads. After that, we propose a simple yet effective method, termed as MASKDISTILL, which predicts the normalized semantic features from CLIP's visual encoder at masked positions based on the corrupted input image. The simple framework beats many previous works with special designs and shows impressive performance across model sizes and tasks. In the future, we would like to explore the proposed method for multimodal pretraining (Wang et al., 2022).

The proposed MASKDISTILL requires an extra teacher model, similar to the tokenizer in BEIT series. Compared with the methods using pixels as targets, the teacher model in MASKDISTILL needs to spend extra time to obtain target features. Meanwhile, we point out that language-guided supervision is not essential in Subsection 4.7 on the academically accessible multi-model datasets, YFCC15M. But whether this conclusion is correct on private 400M image-text pair datasets remains an unknown question.

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

## A Hyperparameters for MaskDistill Pretraining

| Hyperparameters | Base Size | Large Size | Huge Size |
|---|---|---|---|
| Layers | 12 | 24 | 32 |
| Hidden size | 768 | 1024 | 1280 |
| FFN inner hidden size | 3072 | 4096 | 5120 |
| Attention heads | 12 | 16 | 16 |
| Layer scale | 0.1 | 1e-5 | 1e-5 |
| Patch size | $16 \times 16$ | $16 \times 16$ | $14 \times 14$ |
| Training epochs | 300/800 | | |
| Batch size | 2048 | | |
| Adam $\epsilon$ | 1e-8 | | |
| Adam $\beta$ | (0.9, 0.999) | | |
| Peak learning rate | 1.5e-3 | | |
| Minimal learning rate | 1e-5 | | |
| Learning rate schedule | Cosine | | |
| Warmup epochs | 10 | | |
| Stoch. depth | 0.1 | 0.2 | 0.25 |
| Gradient clipping | 3.0 | | |
| Dropout | ✗ | | |
| Weight decay | 0.05 | | |
| Data Augment | RandomResizeAndCrop | | |
| Input resolution | $224 \times 224$ | | |
| Color jitter | 0.4 | | |

Table 11: Hyperparameters for MASKDISTILL pretraining on ImageNet-1K.

## B Hyperparameters for ADE20K Semantic Segmentation Fine-tuning

| Hyperparameters | ViT-B/16 | ViT-L/16 |
|---|---|---|
| Relative positional embeddings | ✓ | |
| Shared relative positional embeddings | ✗ | |
| Peak learning rate | {0.5, 0.8, 1.0, 1.5}e-4 | |
| Fine-tuning steps | 160K | |
| Batch size | 16 | |
| Adam $\epsilon$ | 1e-8 | |
| Adam $\beta$ | (0.9, 0.999) | |
| Layer-wise learning rate decay | 0.75 | 0.85 |
| Minimal learning rate | 0 | |
| Learning rate schedule | Linear | |
| Warmup steps | 1500 | |
| Dropout | ✗ | |
| Stoch. depth | 0.1 | 0.2 |
| Weight decay | 0.05 | |
| Input resolution | $512 \times 512$ | |

Table 12: Hyperparameters for fine-tuning MASKDISTILL on ADE20K.

## C   Hyperparameters for Image Classification Fine-tuning

Table 13: Hyperparameters for fine-tuning MASKDISTILL on ImageNet-1K.

| Hyperparameters | ViT-B/16 | ViT-L/16 | ViT-H/14 |
|---|---|---|---|
| Peak learning rate | 5e-4 | 5e-4 | 2e-4 |
| Fine-tuning epochs | 100 | 50 | 30 |
| Warmup epochs | 20 | 5 | 5 |
| Layer-wise learning rate decay | 0.65 | 0.8 | 0.85 |
| Batch size | | 1024 | |
| Adam $\epsilon$ | | 1e-8 | |
| Adam $\beta$ | | (0.9, 0.999) | |
| Minimal learning rate | | 1e-6 | |
| Learning rate schedule | | Cosine | |
| Stoch. depth | 0.1 | 0.2 | 0.25 |
| Repeated Aug | | ✗ | |
| Weight decay | | 0.05 | |
| Label smoothing $\varepsilon$ | | 0.1 | |
| Dropout | | ✗ | |
| Gradient clipping | | ✗ | |
| Erasing prob. | | 0.25 | |
| Input resolution | | $224 \times 224$ | |
| Rand Augment | | 9/0.5 | |
| Mixup prob. | | 0.8 | |
| Cutmix prob. | | 1.0 | |

