# OpenReview forum: "A Unified View of Masked Image Modeling"
_TMLR — Accepted by TMLR_

### Review · Reviewer_bw3D · 2022-12-02

**Summary Of Contributions:**

The paper proposes to revisit the mask image modeling approach by unifying the approach using an external model and the approach using reconstruction from the original image. The paper evaluates its approach on image classification and semantic segmentation tasks.

**Audience:**

Yes

**Broader Impact Concerns:**

No concerns

**Claims And Evidence:**

No

**Requested Changes:**

At present it is difficult to position the proposed approach with approaches in the literature. Indeed, there are too many changes that interfere with the comparison. The adjustments suggested are in weakness "Comparison with other approaches".
The point "Experiments to show the advantage of mask image modeling" would strengthen the work but is not critical

**Strengths And Weaknesses:**



Strengths:

- The paper is well written and easy to follow.
- The idea seems interesting and the results are quite good.
- There is experiments on semantic segmentation and classification task which is good practice.
- The ablations table Table 6, 7 and 8 are interesting.

Weakness:

- Architecture used for the experiments:

The paper state: "We conduct MIM pretraining on ImageNet-1k (Russakovsky et al., 2015) for base-, large- and huge-size ViTs." according to the code provided in appendix it's seem the architecture is a beit like architecture (for instance with relative position encoding). These are important changes. It is important to clarify that the architecture used is not ViT to avoid confusion.


- Comparison with other approaches:

a) The Table 2 comparison is not very fair as the teacher models used were not pre-trained on the same data.
It would be interesting to merge Table 2 and  Table 6 by mentioning the data used for the teacher and providing results with Maskdistill teacher trained with different dataset to be comparable with other architecture.

b) Table 2 and 4: The paper refers to ViT for all the architecture. However some architecture used relative position encoding and some other not. It is important to clarify this because the comparison is misleading. An interesting comparison would be to use ViT-B as defined in the original paper a teacher trained on ImageNet only and compare the results with MAE or PeCO.

c) Currently there are no comparisons with other approches quite similar like MAE that could be considered as an ablation. Each time there some change in the architecture , change in the initialisation (according to the code the paper used a kind of Fixup `fix_init_weight` in the code) and extra training data  used for the teacher. But there is no ablation on this component.

d) Table 2 The different approaches presented do not always use a teacher. It would be interesting to report the training cost of each approach to understand the different trade-offs.

- Experiments to show the advantage of mask image modeling:

The paper state: "Masked image modeling has demonstrated great potential to eliminate the label-hungry problem of training large-scale vision Transformers, achieving impressive performance on various downstream tasks".

So the ImageNet-1k to ImageNet-1k is maybe to restrictive for self-supervised approaches. It may be interesting to consider the following settings:

a) Pre-training on larger dataset:
There is a lack of experiments with pre-training on a large dataset without labels followed only by a full finetuning on a smaller dataset. For instance a pre-training on ImageNet-1k  followed by full finetuning on ImageNet-1k.

b) Pre-training on dataset with different distribution:
For instance a pre-training on iNaturalist-2021  followed by full finetuning on ImageNet-1k.

---

> ### Author Response · Authors · 2023-01-11
> **Response to the Reviewer bw3D**
>
> We would like to express our sincere gratitude for your time and efforts in reviewing our paper.
>
> *W1. Architecture used for the experiments*
>
> **RE**: To clarify it more clearly, we add a column named "Rel. Pos." in Table 2 of the revised manuscript. Moreover, we conduct an ablation experiment, which abandons those changes, as demonstrated in Table 2 of the revised manuscript.
>
> *W2. Comparison with other approaches*
>
> **RE**: a) As you suggest, we mention "Teacher training dataset" in Table 2 of the revised manuscript.
>
> b) As you suggest, we conduct an ablation experiment, which abandons those changes, as demonstrated in Table 2 of the revised manuscript. Without those changes, the fine-tuning results change from 85.0\% to 84.9\%.
>
> c) According to BEiT, this function is employed to stabilize pretraining and has no impact on the fine-tuning results. Our experimental results also verify it, according to your suggestion.
>
> d) According to your suggestion, we choose some representative methods and measure the training cost, and then report it in Table 6 of the revised manuscript.
>
> *W3. Experiments to show the advantage of mask image modeling*
>
> **RE**: In Table 3 of the revised manuscript, MaskDistill using ViT-H/14 can achieve 88.3\% with 300 epochs pretraining schedule, which is comparable to ViT-H/14 trained on the private JFT-300M. This result has showed the advantage of mask image modeling in terms of the training efficiency and public accessibility. As for the training dataset, paper [1] has conducted extensive experiments on various scale datasets. Despite the gap in specific method, we believe that the tendency still holds for the proposed method.
>
> [1] On Data Scaling in Masked Image Modeling

---

### Review · Reviewer_yZN2 · 2022-12-03

**Summary Of Contributions:**

- The paper proposes a unified view of various masked image modeling-based SSL approaches.
- Specifically, the paper divides them into : teacher models, normalization layer, student model, the MIM head and the loss function used.
- A simple method termed MaskDistill is proposed which trains a student model to reconstruct normalized semantic features from teacher model at masked positions.
- The authors show that the proposed approach leads to consistent improvements on IN top-1 accuracy and downstream task of semantic segmentation.

**Audience:**

Yes

**Broader Impact Concerns:**

A broader impact section is not included. I do not think this is a major concern for this paper though.

**Claims And Evidence:**

Yes

**Requested Changes:**

Please refer to Weaknesses & Questions. (2) and (3) are critical for me. (1), (4) and (5) will help strengthen the paper.
Minor writing comments :
  - avoid vague terms like : Page 1 "proper loss function"
  - It would help to make the captions verbose.
  - Conclusion "susection"

**Strengths And Weaknesses:**

Strengths:

1. The proposed way to unify various MIM approaches seems reasonable and many of the recent approaches in a common setting based on  some of the design decisions.
2. MaskDistill is a simple and effective way to distill information from self-supervised pre-trained teacher models to student models for self-supervised learning.
3. The method seems to lead to consistent improvements in the settings considered.
4. The authors provide a good analysis of the proposed approach

Weaknesses/Questions:
1. Can your approach be extended to non-ViT based teachers ? For example using ResNets which have been explored a lot in the prior SSL work ? One approach could perhaps be use of pre-pooling features as proxy to patch features.
2.  Missing baseline : It seems that an important baseline is missing : Start with the teacher checkpoint and use BEiT/MAE/etc to do SSL on pretraining dataset without any model distillation. This will help evaluate whether distillation is needed
3. The authors mention the extra time & compute for forward pass through the teacher. This could be of practical significance. It will be beneficial to explicitly include this information (training time, FLOP) in the analysis section.
4. Did the authors try to include the distillation loss in addition to the usual MIM loss with separate heads? For example distillation + MAE-like pixel reconstruction
5. It will be interesting to perform some analysis on effect of the proposed technique on transfer learning tasks like [a].

[a] How Well Do Self-Supervised Models Transfer?

---

> ### Author Response · Authors · 2023-01-11
> **Response to the Reviewer yZN2**
>
> We would like to express our sincere gratitude for your time and efforts in reviewing our paper.
>
> *W1. Can your approach be extended to non-ViT based teachers?*
>
> **RE**: In Exp. \#7 in Table 7 of the revised paper, the teacher model is ResNet, which is trained in a supervised way. Despite the gap in architecture, the student still enjoys the significant gain.
>
> *W2. Missing baseline : It seems that an important baseline is missing.*
>
> **RE**: According to your suggestion, we conduct experiments on it. Specifically, we load the public CLIP base model checkpoint as the initialization and perform BEiT pretraining with 300 epochs schedule on ImageNet-1k. We sweep a large lr range (i.e., \{5e-4, 1e-3, 2e-3, 3e-3, 4e-3\}) for a fair comparison. The model obtains 82.6\% top-1 accuracy, which is significantly inferior to MaskDistill.
>
> *W3. It will be beneficial to explicitly include this information (training time, FLOP) in the analysis section.*
>
> **RE**:  We report it in the Table 6 of the revised paper. Compared the proposed MaskDistill with others, MaskDistill enjoys faster training time to achieve the comparable performance.
>
> *W4. Did the authors try to include the distillation loss in addition to the usual MIM loss with separate heads?*
>
> **RE**:  We have not tried to simultaneously reconstruct features and pixel values. To make it work, two MAE-like decoders are required, which seems a little heavy. We'd like to leave the extension for future exploration. Thanks for pointing out the potential way for improvement.

---

> > ### Comment · Reviewer_yZN2 · 2023-01-16
> > **Thanks for the response**
> >
> > Thanks for the response. The responses to W1-W4 adequately address my concerns.
> > Do you have any follow up experiments for transfer learning (W5) ?

---

> > > ### Author Response · Authors · 2023-01-17
> > > **Response to W5**
> > >
> > > As concluded in the paper[a], the ImageNet Top-1 accuracy is correlated with transfer to many-shot recognition, few-shot, object detection and dense prediction.
> > > In fact, we have conducted the dense prediction task (semantic segmentation on ADE20K) and compared results in Table 2 and 7 in the paper.
> > > In Table 2, the comparison results indicate that higher ImageNet Top-1 accuracy tends to be correlated with higher segmentation performance.
> > > In Table 7, the ablation results also support this conclusion.
> > >
> > > [a] How Well Do Self-Supervised Models Transfer?

---

> > > > ### Comment · Reviewer_yZN2 · 2023-01-19
> > > > **Thanks for your response**
> > > >
> > > > Thanks for your response.

---

### Review · Reviewer_FqMf · 2022-12-21

**Summary Of Contributions:**

This paper focuses on the problem of masked image modeling. The author proposes a unified view of masked image modeling through the combination of a teacher model, a normalization layer, a student model, a MIM head, and a proper loss function. The simple yet effective method called MaskDistill is based on the unified view of MIM methods. Extensive experimental evaluations are conducted on several downstream tasks, and the results seem good under various settings.

**Audience:**

Yes

**Broader Impact Concerns:**

This paper does not involve ethical issues.

**Claims And Evidence:**

Yes

**Requested Changes:**

The writing is clear and the content is easy to follow. No additional revision is needed from my perspective.

**Strengths And Weaknesses:**

Strengths:
- The paper concludes the existing MIM methods into a unified view and optimizing existing settings in a simple and clean framework. Especially, it is nice that Table 1 and Table 2 clearly illustrate the difference between existing methods and the proposed MaskDistill.
- As an experiment-driven model design, extensive experiments are conducted to prove the effectiveness of the module selection of the unified MIM framework. The comparison analysis with BEiT v2 by codebook collapse is also interesting.

Weaknesses:
- This work is a good successor to past MIM methods, but it has not made further breakthrough contributions to the current MIM framework. Its technical contribution is limited to empirical influence on the research area. Readers are hard to be inspired with limited insights.

---

> ### Author Response · Authors · 2023-01-11
> **Response to Reviewer FqMf**
>
> We would like to express our sincere gratitude for your time and efforts in reviewing our paper. It is great to see that you recognize the strengths of our paper, including the unified view of existing MIM methods and the clear illustration of the differences between existing methods and the proposed MaskDistill. We also recognize that the contribution is somewhat limited to empirical influence on the research area.
> We believe that we proposed a powerful and comprehensive framework to boost a model that trained from any methods, as demonstrated in Table 7 in the revised paper. That is, we can boost the performance of the teacher models in various downstream tasks by using MaskDistill framework, which is also our contribution and could inspire some readers.

---

### Decision · Action_Editors · 2023-02-14

**Recommendation:** Accept with minor revision

**Comment:**

 While the novelty of the proposed techniques is not very high, reviewers find the paper interesting based on the strong experimental section and results. The paper brings in new results to an area with strong current research interest. I recommend a minor revision since two of the reviewers think the authors should incorporate an additional downstream task and adjust the manuscript further based on their comments (see recommendation from reviewer FqMf).

**Audience:**

Masked image modelling and self-supervised learning are popular research areas where the results of this paper will be relevant.
I believe it is of sufficient interest.

**Claims And Evidence:**

The paper proposes an interesting synthesis of existing masked image modelling approaches and provides a good experimental evaluation.

All reviewers are satisfied with the existing results in the paper -- "experimental evaluations demonstrate the effectiveness of the proposed approach", "the improvements are consistent".

One of them thinks that additional transfer learning tasks would strengthen the paper and I encourage the authors to add an extra experiment in a minor revision.